# Cyclosporine may reduce the risk of symptomatic COVID-19 in patients with systemic lupus erythematosus: a retrospective cohort study

He-Jun Li,[1,2] Shu-Huan Lin,[2] Yan-Qing Wang,[2] Ling Chen,[2] Xiang-Xiong Zheng,[2] Li-Xin Wei[1,3]

**ABSTRACT** This study aimed to explore the effect of cyclosporine (CsA) on severe acute respiratory syndrome coronavirus 2 (SARS-CoV-2) infection in systemic lupus erythematosus (SLE) patients to provide a valuable reference for clinical treatment strategies in the context of the long-term risk of SARS-CoV-2 infection. SLE patients who visited the Rheumatology Outpatient Department of Fujian Medical University Union Hospital between 1 May and 31 October 2022 were included. Data on SARS-CoV-2 infection in patients between 1 November 2022 and 31 July 2023 were obtained by telephone follow-up. Patients were divided into two groups according to whether CsA was used during the observation period: the glucocorticoid or hydroxychloroquine group and the CsA group. To assess the robustness of the results, Data sets 1 and 2 were established to be analyzed independently. Multivariate logistic regression was used to estimate the odds ratios (ORs) and 95% confidence intervals (CIs) for symptomatic coronavirus disease (COVID-19). A total of 184 patients were included, among whom 129 were definite symptomatic COVID-19 patients; 29 were presumptive symptomatic COVID-19 patients; and 4 had signs and symptoms of COVID-19, but tested negative for SARS-CoV-2 in a virological test. According to the multivariable-adjusted models, CsA was associated with lower odds of symptomatic COVID-19 ($P = 0.042$, OR = 0.316, 95% CI: 0.104–0.959 in Data set 1 and $P = 0.021$, OR = 0.257, 95% CI: 0.081–0.812 in Data set 2). CsA is associated with lower odds of contracting symptomatic COVID-19. The use of CsA may be considered an appropriate therapeutic option for disease management in patients with rheumatic diseases who have severe disease activity and persistent SARS-CoV-2 infection.

**IMPORTANCE** Our study indicated that cyclosporine may reduce the risk of symptomatic COVID-19 in systemic lupus erythematosus patients in spite of its immunosuppressive effects. This study provides a reference for clinical treatment strategies for AIIRD patients in the context of the long-term risk of SARS-CoV-2 infection.

**KEYWORDS** systemic lupus erythematosus, cyclosporine, immunosuppressive therapy, SARS-CoV-2, COVID-19

The coronavirus disease (COVID-19) pandemic, which was caused by severe acute respiratory syndrome coronavirus 2 (SARS-CoV-2), has had a global effect on public health. Due to the strong infectivity of SARS-CoV-2 and the short duration of immunity after infection, repeated infection with SARS-CoV-2 may be a long-term public health problem. Vaccination against SARS-CoV-2 reduces symptomatic COVID-19 infection rates and poor outcomes (1). Therefore, SARS-CoV-2 infection has little impact on the general population. However, vaccination has additional considerations for people with autoimmune inflammatory rheumatic diseases (AIIRD), including sub-optimal vaccine responses caused by immunomodulatory drugs and rheumatic disease flares (2, 3).

Address correspondence to He-Jun Li, tanklhj@163.com, or Li-Xin Wei, drwei234@163.com.

He-Jun Li and Shu-Huan Lin contributed equally to this article. Author order was determined by drawing straws.

The authors declare no conflict of interest.

Moreover, the general recommendation is to postpone vaccination in AIIRD patients with disease activity. Consequently, the overall vaccination rate and effective rate in AIIRD patients are much lower than those in the general population. In addition, given that immunomodulatory drugs widely used in AIIRD patients may be associated with a moderate increase in infection risk, SARS-CoV-2 infection remains a long-standing challenge for AIIRD patients. In the context of the long-term risk of SARS-CoV-2 infection, it is of significant clinical value to explore the impact of various immunomodulatory drugs on the risk of contracting COVID-19 in AIIRD patients.

Cyclosporine (CsA) is an immunomodulatory drug commonly used to treat systemic lupus erythematosus (SLE) that can inhibit the activation of T cells. Although its immunosuppressive effects can lead to delayed clearance of the virus, CsA has antiviral effects on cell lines cultured *in vitro* (4, 5). Thus, CsA has dual effects (both beneficial and harmful) on SARS-CoV-2 infection. It is particularly instructive for clinical practice to explore the influence of the dual effects of CsA on SARS-CoV-2 infection in AIIRD patients.

The risk of SARS-CoV-2 infection in AIIRD patients has been controversial in different studies (6, 7). The heterogeneity in these results might reflect differences in disease, current levels of disease activity, and specific disease-related comorbidities. Moreover, differences in the use of glucocorticoids (GCs) and/or immunomodulatory drugs across studies, both of which are risk factors for serious infection (8), are another important reason for the inconsistent results. Thus, many confounding factors need to be controlled to explore the real impact of a drug on SARS-CoV-2 infection.

Therefore, this study aimed to explore the effect of CsA on SARS-CoV-2 infection in SLE patients after controlling for confounders to provide a valuable reference for clinical treatment strategies.

## MATERIALS AND METHODS

### Study cohort and patients

This was a retrospective cohort study of the Chinese Han population. SLE patients who visited the Rheumatology Outpatient Department of Fujian Medical University Union Hospital from 1 May to 31 October 2022 were included. The medical records were retrospectively reviewed, and patients were followed through telephone calls for observation in August 2023. The study included patients who were 18–75 years of age, inclusive, at the time of screening. The eligibility criteria were as follows: (1) fulfilled either the 2012 Systemic Lupus International Collaborating Clinics (SLICC) Classification Criteria (9) or the 2019 EULAR/ACR Classification Criteria for SLE (10); (2) had a daily maintenance dose of prednisone (or equivalent) ≤10 mg during the observation period; (3) received no other immunomodulatory drugs, except for GCs, hydroxychloroquine (HCQ), or CsA during the period from 1 May to 31 October 2022 and did not change to different immunomodulatory drugs during the observation period; (4) were not pregnant; (5) did not have COVID-19 before 31 October 2022; and (6) had no irreversible damage to important organs. Patients who were lost to telephone follow-up, uncertain about infection, or lacked important data were excluded.

Based on whether CsA was used during the observation period, patients were divided into two groups: the GC or HCQ group and the CsA group.

### Collection and definition of data

We obtained demographic and clinical data using a review of electronic medical records and received confirmation from patients during telephone follow-up. We analyzed the following parameters: sex, age, clinical manifestations, and rheumatic disease medications taken immediately before infection. Information on SLE-related organ involvement, such as cutaneous, arthritis, serositis, lupus nephritis (LN), neuropsychiatric SLE (NPSLE), and hematological involvement (leukopenia, thrombocytopenia, and hemolytic anemia), was collected from the medical records of all included patients.

The study period was from 1 November 2022 (when the level of COVID-19 prevention and control in China was gradually lowered) to 31 July 2023, and the primary outcome measure was symptomatic COVID-19. Data on SARS-CoV-2 infection among patients were collected through telephone follow-up during the study period. Based on virological detection, SLE patients who had any of the various signs and symptoms of COVID-19, such as fever, cough, sore throat, malaise, headache, muscle pain, nausea, vomiting, diarrhea, and loss of taste and smell, during the study period were categorized as follows: (1) definite symptomatic COVID-19 patients who tested positive for SARS-CoV-2 using a virological test, that is, a nucleic acid amplification test or an antigen test, during the study period; (2) presumptive symptomatic COVID-19 patients who had epidemiological evidence but did not test for virology during the pandemic (the period from 1 November 2022 to 28 February 2023); and (3) non-COVID-19 patients who yielded negative results for SARS-CoV-2 through virological testing.

SLE patients without COVID-19-related symptoms were defined as individuals who had no symptoms consistent with COVID-19 during the study period, including individuals who tested positive for SARS-CoV-2 using a virological test, individuals who yielded negative results for SARS-CoV-2 through virological testing, and individuals who had not undergone virus testing.

To assess the robustness of the results, we established two datasets—Data sets 1 and 2—for analysis. In Data set 1, symptomatic COVID-19 patients included definite and presumptive symptomatic COVID-19 patients, while non-symptomatic COVID-19 patients included non-COVID-19 patients and SLE patients without COVID-19-related symptoms. In Data set 2, symptomatic COVID-19 patients were referred to definite symptomatic COVID-19 cases, while non-symptomatic COVID-19 patients were referred to SLE patients without COVID-19-related symptoms.

Disease duration was defined as the time interval from diagnosis of SLE to 31 October 2022. The observation duration was defined as the time interval between 1 November 2022 and the initial manifestation of any signs or symptoms of COVID-19 for symptomatic patients or until 31 July 2023 for SLE patients without COVID-19-related symptoms.

## Statistical analysis

Categorical variables were described as numbers (percentages) and compared using the Chi-square test or Fisher's exact test between the GC or HCQ group and the CsA group. Normally distributed continuous data were presented as the mean and standard deviation (SD), and differences between groups were tested by one-way analysis of variance. Abnormally distributed quantitative variables were expressed as medians (interquartile range [IQR]) and compared between groups using the Mann−Whitney nonparametric $U$ test.

The associations between the risk of symptomatic COVID-19 and baseline characteristics were analyzed by a univariate logistic regression model in Data sets 1 and 2. To further investigate the effect of CsA on the risk of symptomatic COVID-19, each variable with significant differences ($P < 0.05$) in the univariate logistic regression analysis was included in the multivariable logistic regression model to account for potential confounding factors. All reported $P$ values were two-tailed, and statistical significance was defined as $P < 0.05$. All the statistical analyses were performed using R version 4.3.2.

## RESULTS

### Study populations

A total of 215 patients were initially enrolled, among whom 18 and four patients were lost to telephone follow-up in the GC + HCQ group and the CsA group, respectively. In addition, four patients were excluded for missing important data, and five patients were excluded for uncertain primary outcomes. Ultimately, 184 patients were included in the analysis. As shown in Table 1, of the 184 patients, 162 patients had one or more signs and symptoms of COVID-19 during the study period. Of these 162 patients, 129 were

**TABLE 1** Infection and manifestations of SARS-CoV-2 in patients with SLE[a]

| | Patients with any of the various signs and symptoms of COVID-19 (n = 162) | | | SLE patients without COVID-19-related symptoms (n = 22) | |
|---|---|---|---|---|---|
| | Definite symptomatic COVID-19 (n = 129) | Presumptive symptomatic COVID-19 (n = 29) | Non-symptomatic COVID-19 (n = 4) | Patients with a positive virological test (n = 2) | Patients without a positive virological test (n = 20) |
| GC + HCQ group, n (%) | 116 (89.9) | 25 (86.2) | 2 (50) | 2 (100) | 14 (70) |
| CsA group, n (%) | 13 (10.1) | 4 (13.8) | 2 (50) | 0 | 6 (30) |
| Fever, n (%) | 152 (95.6) | 21 (84.0) | 0 | 0 | 0 |
| Pneumonia, n (%) | 2 | 0 | 0 | 0 | 0 |
| Death, n (%) | 0 | 0 | 0 | 0 | 0 |

[a]GCs: glucocorticoids; HCQ: hydroxychloroquine; CsA: cyclosporine.

definite symptomatic COVID-19 patients; 29 were presumptive symptomatic COVID-19 patients; and four were non-COVID-19 patients. There were two cases of pneumonia, but no fatalities occurred. The total number of SLE patients without COVID-19-related symptoms was 22, among whom two tested positive for SARS-CoV-2.

## Comparison of clinical characteristics between the GC or HCQ group and the CsA group

There was no statistically significant difference in age between the two groups (years, mean ± SD, 39.6 ± 12.7 versus 38.2 ± 11.2 in Data set 1 and 39.1 ± 12.3 versus 38.2 ± 10.9 in Data set 2, $P > 0.05$). The median observation duration was 2 months (IQR 2–3) in the GC or HCQ group and 3 months (IQR 2–9) in the CsA group either in Data set 1 or 2. In both data sets, the proportion of people with LN was significantly greater in the CsA group than in the GC or HCQ group (64.0% versus 21.4% in Data set 1 and 68.4% versus 23.5% in Data set 2, $P < 0.001$), while there was no statistically significant difference in other SLE-related organ involvement ($P < 0.05$). The proportion of people with hypertension was greater in the CsA group than in the GC or HCQ group (32.0% versus 12.6% in Data set 1 and 31.6% versus 9.1% in Data set 2, $P < 0.05$). The vaccination rates were greater than 60% in both groups, and there was no statistically significant difference between the two groups. Compared with those in the GC and HCQ groups, the dose of GCs was slightly greater ($P < 0.05$), while the dose of HCQ tended to decrease in the CsA group ($P = 0.011$ in Data set 1 and $P = 0.064$ in Data set 2). In terms of SARS-CoV-2 infection, the incidence of symptomatic COVID-19 was inclined to decline in the CsA group than in the GC or HCQ group (68% versus 88.7%, $P = 0.011$ in Data set 1 and 68.4% versus 87.9%, $P = 0.064$ in Data set 2). The details are shown in Table 2.

## The effect of CsA on the occurrence of symptomatic COVID-19

Univariate logistic regression analysis revealed that CsA ($P = 0.009$, odds ratio [OR]= 0.271, 95% confidence interval [CI]: 0.103–0.718 in Data set 1 and $P = 0.031$, OR = 0.299, 95% CI: 0.100–0.897 in Data set 2) reduced the risk of symptomatic COVID-19, and the risk of symptomatic COVID-19 decreased slightly with increasing age ($P = 0.012$, OR = 0.958, 95% CI: 0.926–0.991 in Data set 1 and $P = 0.013$, OR = 0.956, 95% CI: 0.915–0.989 in Data set 2) (Fig. 1 and 2). HCQ increased the risk of symptomatic COVID-19 in Data set 1 ($P = 0.007$, OR = 2.501, 95% CI: 1.288–4.856) (Fig. 1), while there was no statistically significant effect in Data set 2 ($P = 0.053$, OR = 2.183, 95% CI: 0.990–4.817) (Fig. 2). The details are shown in Fig. 1 and 2. After adjusting for confounding factors, CsA still showed an effect in reducing the risk of symptomatic COVID-19 ($P = 0.042$, OR = 0.316, 95% CI: 0.104–0.959 in Data set 1 and $P = 0.021$, OR = 0.257, 95% CI: 0.081–0.812 in Data set 2) according to the multivariable logistic regression model (Fig. 3). In addition, the multivariable logistic regression model also showed a slight decrease in the risk of symptomatic COVID-19 with increasing age ($P = 0.011$, OR = 0.954, 95% CI: 0.919–0.989 in Data set 1 and $P = 0.009$, OR = 0.947, 95% CI: 0.909–0.987 in Data set 2) (Fig. 3).

**TABLE 2** Comparison of clinical characteristics between GC or HCQ and CsA groups by bivariate analysis in data sets 1 and 2[a,b,c,d]

| | Data set | | | Data set | | |
|---|---|---|---|---|---|---|
| | GC or HCQ group (n = 159) | CsA group (n = 25) | P value | GC or HCQ group (n = 132) | CsA group (n = 19) | P value |
| Gender, F (%) | 152 (95.6) | 21 (84.0) | 0.069 | 126 (95.5) | 16 (84.2) | 0.156 |
| Age, years, mean ± SD | 39.6 ± 12.7 | 38.2 ± 11.2 | 0.603 | 39.1 ± 12.3 | 38.2 ± 10.9 | 0.773 |
| Disease duration, years, mean ± SD | 8.5 ± 5.7 | 10.3 ± 6.4 | 0.147 | 8.5 ± 5.7 | 9.8 ± 6.7 | 0.375 |
| Observation duration, months, median [IQR] | 2 [2,3] | 3 [2,9] | 0.094 | 2 [2,3] | 3 [2,9] | 0.052 |
| Mucocutaneous manifestations, n (%) | 97 (61.0) | 11 (44.0) | 0.108 | 78 (59.1) | 8 (42.1) | 0.162 |
| Arthritis, n (%) | 68 (42.8) | 6 (24.0) | 0.075 | 60 (45.5) | 5 (26.3) | 0.115 |
| Lupus nephritis, n (%) | 34 (21.4) | 16 (64.0) | <0.001 | 31 (23.5) | 13 (68.4) | <0.001 |
| NPSLE, n (%) | 5 (3.1) | 1 (4.0) | 0.589 | 3 (2.3) | 1 (5.3) | 0.419 |
| Hemolytic anemia, n (%) | 9 (5.7) | 1 (4.0) | 1.000 | 8 (6.1) | 0 | 0.553 |
| Leukopenia, n (%) | 38 (23.9) | 9 (36.0) | 0.197 | 29 (22.0) | 8 (42.1) | 0.105 |
| Thrombocytopenia, n (%) | 49 (30.8) | 6 (24.0) | 0.489 | 38 (28.8) | 3 (15.8) | 0.234 |
| Serositis, n (%) | 11 (6.9) | 1 (4.0) | 0.910 | 9 (6.8) | 1 (5.3) | 1.000 |
| cSLEDAI-2K, n (%) | 0 [0,0] | 0 [0, 0] | 0.214 | 0 [0,0] | 0 [0,0] | 0.119 |
| BMI, median [IQR] | 21.5 [19.9,22.9] | 21.8 [19.2, 23.6] | 0.862 | 21.4 [19.8, 22.5] | 21.6 [18.7, 23.4] | 0.996 |
| Hypertension, n (%) | 20 (12.6) | 8 (32.0) | 0.027 | 12 (9.1) | 6 (31.6) | 0.014 |
| Diabetes, n (%) | 2 (1.3) | 1 (4) | 0.356 | 1 (0.8) | 1 (5.3) | 0.237 |
| Coronary heart disease, n (%) | 0 | 1 (4.0) | 0.136 | 0 | 0 | |
| COVID-19 vaccination, n (%) | 118 (74.2) | 17 (68.0) | 0.514 | 99 (75.0) | 12 (63.2) | 0.274 |
| Dose of GCs, mg/d, median [IQR] | 2.5 [2.5, 5.0] | 5.0 [5.0, 5.0] | <0.001 | 2.5 [2.5, 5.0] | 5.0 [5.0, 5.0] | <0.001 |
| Dose of HCQ, g/d, median [IQR] | 0.2 [0.2, 0.2] | 0.2 [0.1, 0.2] | 0.011 | 0.2 [0.2, 0.2] | 0.2 [0.2, 0.2] | 0.064 |
| Dose of CsA, mg/d, median [IQR] | 0 [0, 0] | 150 [100, 150] | <0.001 | 0 [0,0] | 150 [100, 150] | <0.001 |
| Symptomatic COVID-19, n (%) | 141 (88.7) | 17 (68.0) | 0.014 | 116 (87.9) | 13 (68.4) | 0.057 |

[a]Disease duration: the time interval from diagnosis of SLE to 31 October 2022.
[b]Observation duration: the time interval from the time interval between 1 November 2022 and the initial manifestation of any signs or symptoms of COVID-19 for symptomatic patients, or until 31 July 2023 for SLE patients without COVID-19-related symptoms.
[c]The dose of GCs, HCQ, and CsA referred to the dose taken by patients immediately prior to infection.
[d]NPSLE: neuropsychiatric systemic lupus erythematosus; GCs: glucocorticoids (prednisone; HCQ: hydroxychloroquine; CsA: cyclosporine).

## DISCUSSION

Our study focused on symptomatic COVID-19 as the main endpoint and revealed the impact of CsA on the risk of symptomatic COVID-19 in patients with SLE. To our knowledge, this is the first study to show that CsA may reduce the risk of symptomatic COVID-19 in patients with SLE. Our findings remained largely unchanged, excluding those with a presumptive diagnosis (n = 29) and those with symptoms, but a negative virological test (n = 4).

Previous studies have focused mainly on the association of immunomodulators with COVID-19-related hospitalization or death (11, 12). It is difficult to obtain accurate data on non-hospitalized COVID-19 patients, as a significant portion of them may remain undocumented. Furthermore, the absence of virological testing led to the underidentification of some presumptive symptomatic COVID-19 cases. The period from November 2022 to February 2023 was the transition stage of China's COVID-19 prevention and control strategy, during which the infection rate of SARS-CoV-2 sharply increased. The short observation duration in our study suggested that most people were infected with SARS-CoV-2 during this period. Meanwhile, after several years of adequate preparation, there was sufficient capacity for virological testing, either nucleic acid amplification by professional institutions or simple antigen testing, to meet the diagnostic requirements of patients. Consequently, most patients (88.0%, 162 out of 184) had one or more signs and symptoms of COVID-19 during the study period, and of these symptomatic patients, the majority (82.1%, 133 out of 162) were tested for SARS-CoV-2 using a virological test, which objectively provided a source of cases with a definite diagnosis for our research. There is, of course, one further point to make. Despite adequate virological test

| Variables | Level | OR | 95%CI | | P value |
|---|---|---|---|---|---|
| Female | Yes vs No | 1.380 | 0.281–6.777 | | 0.692 |
| Age,years | | 0.958 | 0.926–0.991 | | 0.012 |
| Mucocutaneous manifestations | Yes vs No | 1.806 | 0.784–4.162 | | 0.165 |
| Arthritis | Yes vs No | 1.320 | 0.554–3.144 | | 0.530 |
| Lupus nephritis | Yes vs No | 0.542 | 0.228–1.292 | | 0.167 |
| NPSLE | Yes vs No | 0.817 | 0.092–7.287 | | 0.856 |
| Hemolytic anemia | Yes vs No | 1.510 | 0.183–12.442 | | 0.702 |
| Leukopenia | Yes vs No | 0.920 | 0.360–2.350 | | 0.862 |
| Thrombocytopenia | Yes vs No | 0.637 | 0.269–1.509 | | 0.306 |
| Serositis | Yes vs No | 0.811 | 0.167–3.930 | | 0.795 |
| cSLEDAI−2K | | 0.554 | 0.287–1.069 | | 0.078 |
| Body Mass Index | | 1.029 | 0.879–1.204 | | 0.725 |
| Hypertension | Yes vs No | 2.404 | 0.901–6.409 | | 0.080 |
| COVID−19 vaccination | Yes vs No | 1.018 | 0.399–2.594 | | 0.971 |
| Dose of GCs,mg/d | | 0.945 | 0.772–1.155 | | 0.578 |
| Dose of HCQ,0.1g/d | | 2.501 | 1.288–4.856 | | 0.007 |
| CsA | Yes vs No | 0.271 | 0.103–0.718 | | 0.009 |

FIG 1 Risk factors for symptomatic COVID-19 in Data set 1 according to univariate logistic regression analysis. The figure presents the ORs and 95% CIs associated with the endpoint. OR: odds ratio. See Table 2 note for expansion of additional abbreviations.

capabilities, 17.9% (29/162) of the enrolled patients with symptoms were not tested for SARS-CoV-2 using a virological test. Moreover, it is important to note that no virological test can achieve a 100% positive rate, indicating that the possibility of SARS-CoV-2 infection cannot be completely ruled out, even with a negative virological test result for SARS-CoV-2. Therefore, to minimize bias and enhance the reliability of our findings, we established two separate data sets for analysis—Data set 1 and Data set 2—based on virological detection.

| Variables | Level | OR | 95%CI | | P value |
|---|---|---|---|---|---|
| Female | Yes vs No | 0.720 | 0.086–6.060 | | 0.763 |
| Age,years | | 0.956 | 0.915–0.989 | | 0.013 |
| Mucocutaneous manifestations | Yes vs No | 2.139 | 0.853–5.366 | | 0.105 |
| Arthritis | Yes vs No | 1.108 | 0.442–2.776 | | 0.827 |
| Lupus nephritis | Yes vs No | 0.677 | 0.262–1.752 | | 0.422 |
| NPSLE | Yes vs No | 0.500 | 0.050–5.036 | | 0.556 |
| Hemolytic anemia | Yes vs No | 1.205 | 0.141–10.301 | | 0.865 |
| Leukopenia | Yes vs No | 1.122 | 0.383–3.284 | | 0.834 |
| Thrombocytopenia | Yes vs No | 0.767 | 0.288–2.041 | | 0.595 |
| Serositis | Yes vs No | 0.661 | 0.131–3.341 | | 0.617 |
| cSLEDAI−2K | | 0.546 | 0.280–1.065 | | 0.076 |
| Body Mass Index | | 1.097 | 0.908–1.325 | | 0.339 |
| Hypertension | Yes vs No | 0.381 | 0.121–1.204 | | 0.100 |
| COVID−19 vaccination | Yes vs No | 1.358 | 0.509–3.619 | | 0.541 |
| Dose of GCs,mg/d | | 0.936 | 0.756–1.159 | | 0.544 |
| Dose of HCQ,0.1g/d | | 2.183 | 0.990–4.817 | | 0.053 |
| CsA | Yes vs No | 0.299 | 0.100–0.897 | | 0.031 |

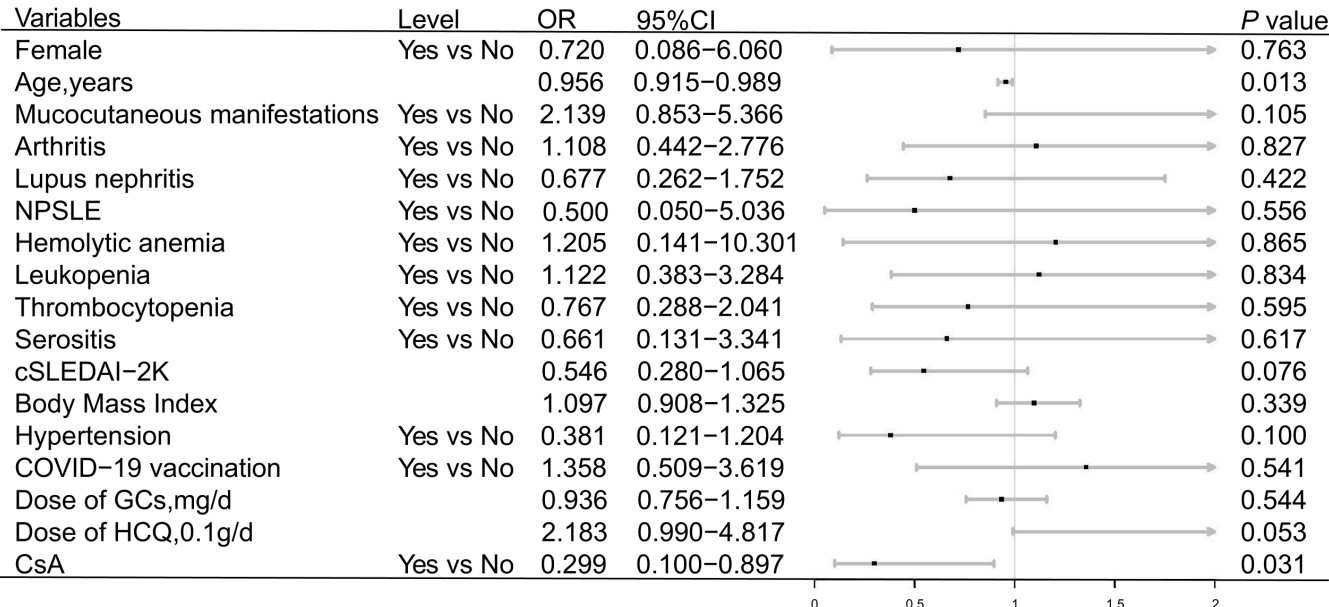

FIG 2 Univariate logistic regression analysis of the risk factors for symptomatic COVID-19 in Data set 2. The figure presents the ORs and 95% CIs associated with the end point. OR: odds ratio. See Table 2 for the expansion of additional abbreviations.

**A**

| Variables | Level | OR | 95%CI | | P value |
|---|---|---|---|---|---|
| Age, years | | 0.954 | 0.919-0.989 | | 0.011 |
| Dose of HCQ, 0.1g/d | | 1.961 | 0.945-4.067 | | 0.070 |
| CsA | Yes vs No | 0.316 | 0.104-0.959 | | 0.042 |

0  0.5  1  1.5  2  2.5  3  3.5  4

**B**

| Variables | Level | OR | 95%CI | | P value |
|---|---|---|---|---|---|
| Age,years | | 0.947 | 0.909-0.987 | | 0.009 |
| CsA | Yes vs No | 0.257 | 0.081-0.812 | | 0.021 |

0  0.25  0.5  0.75  1

**FIG 3** Multivariate logistic regression analysis of the effect of CsA on the occurrence of symptomatic COVID-19. The ORs and 95% CIs are associated with the end point in (A) Data set 1 and (B) Data set 2. OR: odds ratio. See Table 2 note for the expansion of additional abbreviations.

In general, the risk of SARS-CoV-2 infection in people with AIIRDs is affected by immunomodulators, age, disease status, etc. Prednisone doses ≥10 mg/day, rituximab, and moderate/high disease activity may increase the risk of COVID-19-related hospitalization (11, 13). Therefore, to minimize confounders, a prednisone dose ≤10 mg/day was used for patients included in this study, and patients receiving immunomodulatory drugs other than GCs, HCQ, or CsA were excluded. CsA is one of the main drugs used to treat LN, which explains the difference in the proportion of SLE patients with LN and hypertension between the CsA group and the GC or HCQ group. Moreover, perhaps due to the increased proportion of LN patients in the CsA group, the prednisone dose in this group was slightly greater than that in the GC or HCQ group. Despite these differences between the two groups, LN, hypertension, and prednisone dose did not significantly affect the risk of symptomatic COVID-19 according to the univariate logistic regression model ($P > 0.05$), either in Data sets 1 or 2.

SARS-CoV-2 enters cells by binding to angiotensin-converting enzyme 2 (ACE2) through spike proteins and fusing with cell membranes via cleavage of the serine protease TMPRSS2 (14). CsA can inhibit the expression of ACE2 in liver cells (15), and molecular docking and all-atom molecular dynamics simulation results revealed strong and stable binding of cyclosporine A to the TMPRSS2 gene (16). As a result, CsA may inhibit the pathway by which SARS-CoV-2 enters cells. In addition, cyclophilin A is involved in viral replication (17), and CsA can bind to cyclophilin A and may affect viral replication in the cell. *In vitro*, CsA dampened viral infection and cytokine release from lung cells upon exposure to three different SARS-CoV-2 variants (4). In this study, multivariate logistic regression models for both data sets 1 and 2 showed a reduction in the risk of symptomatic COVID-19 in SLE patients treated with CsA, showing the protective effect of CsA against SARS-CoV-2 *in vivo*. The consistency of the analysis results between the two datasets indicates the reliability of the research results. Immunosuppressants are commonly used to treat SLE, especially LN. However, the suppression of immunity may lead to a variety of infections in SLE patients. Moreover, SARS-CoV-2 infection may also lead to SLE flares (18). Therefore, in the context of SARS-CoV-2 infection, there is a dilemma regarding the selection of immunosuppressants for controlling disease activity in patients with SLE. Perhaps, due to its potential anti-SARS-CoV-2 effect, CsA could be considered an appropriate treatment option for

disease control in this specific scenario, particularly for individuals with severe AIIRD and persistent SARS-CoV-2 infection.

There have been many studies on the role of HCQ in the treatment of COVID-19. One study found that treatment with HCQ was associated with a reduction in COVID-19-associated mortality (19), while another did not (20). Post-exposure therapy with HCQ did not prevent SARS-CoV-2 infection or symptomatic COVID-19 in healthy persons exposed to a PCR-positive patient (21). In our study, a univariate regression model in Data set 1 but not in Data set 2 showed that an increase in HCQ dose was associated with the risk of symptomatic COVID-19, indicating that HCQ acted as a facilitator rather than a protector against SARS-CoV-2 infection. However, the multivariable logistic regression model showed no significant association between HCQ and the risk of symptomatic COVID-19 ($P > 0.05$). The observed phenomenon may be attributed to the protective effect of CsA against SARS-CoV-2, as the dosage of HCQ was lower in the CsA group than in the GC and HCQ groups in Data set 1.

Generally, COVID-19 is more severe in patients older than 65 years. Our results showed that increasing age was associated with a low risk of symptomatic COVID-19. Considering the low average age (less than 40 years) in our study, this approach cannot be applied to the general population.

There are limitations that should be noted in the interpretation of our results. The collection of retrospective data is subject to recall bias. To reduce recall bias, we inquired not only about the patient's infection status but also about the infection status of cohabitants to verify each other during the investigation. Besides, it is a single-center study with a small sample size, which could have introduced selection bias.

In conclusion, although CsA has immunosuppressive effects, it may reduce the risk of symptomatic COVID-19 in SLE patients. The use of CsA may be considered an appropriate therapeutic option for disease management in patients with AIIRDs who have severe disease activity and persistent SARS-CoV-2 infection. This study provides a reference for clinical treatment strategies for AIIRD patients in the context of the long-term risk of SARS-CoV-2 infection.

## ACKNOWLEDGMENTS

We thank all the patients and their families involved in this study.

## AUTHOR AFFILIATIONS

[1]Fujian Institute of Clinical Immunology, Fujian Medical University Union Hospital, Fuzhou, China
[2]Department of Rheumatology, Fujian Medical University Union Hospital, Fuzhou, China
[3]Department of Nephrology, Fujian Medical University Union Hospital, Fuzhou, China

## AUTHOR ORCIDs

He-Jun Li  http://orcid.org/0000-0001-9179-8887
Li-Xin Wei  http://orcid.org/0000-0003-1078-9087

## DATA AVAILABILITY

Data are available upon request.

## ETHICS APPROVAL

The cohort study received full ethical approval from the ethics board of Fujian Medical University Union Hospital under project number 2023KY230. Since the study was an observational study conducted through telephone interviews, verbal informed consent was obtained rather than written informed consent, and general confidentiality principles were abided by.

## ADDITIONAL FILES

The following material is available online.

### Open Peer Review

**PEER REVIEW HISTORY (review-history.pdf).** An accounting of the reviewer comments and feedback.

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
