## [Reviewer comments · Microbiology Spectrum]

Microbiology Spectrum

Cyclosporine may Reduce the Risk of Symptomatic COVID-19 in Patients with Systemic Lupus Erythematosus: a Retrospective Cohort Study

He-Jun Li, Shu-Huan Lin, Yan-Qing Wang, Ling Chen, Xiang-Xiong Zheng, and Li-Xin Wei

Corresponding Author(s): He-Jun Li, Fujian Medical University Union Hospital

Review Timeline:

Submission Date:	May 24, 2024
Editorial Decision:	August 20, 2024
Revision Received:	September 14, 2024
Accepted:	September 24, 2024

Editor: Dhammika Navarathna

Reviewer(s): Disclosure of reviewer identity is with reference to reviewer comments included in decision letter(s). The following individuals involved in review of your submission have agreed to reveal their identity: Ruth Chingcuangco Abanador (Reviewer #1); Ghulam Abbas (Reviewer #2)

Transaction Report:

DOI: <https://doi.org/10.1128/spectrum.01276-24>

Re: Spectrum01276-24 (Cyclosporine may Reduce the Risk of Symptomatic COVID-19 in Patients with Systemic Lupus Erythematosus: a Retrospective Cohort Study)

Dear Dr. He-Jun Li:

Thank you for the privilege of reviewing your work. Below you will find my comments, instructions from the Spectrum editorial office, and the reviewer comments.

Revision Guidelines

Sincerely,
Dharmika Navarathna
Editor
Microbiology Spectrum

Reviewer #1 (Comments for the Author):

The authors appropriately and anonymously declared the number of included/excluded patients in the study. However, please explain clearly how COVID-19 asymptomatic patients without a virological test were recruited for this study, since your statement on Page 5 Para. 3 was confusing (Asymptomatic patients were defined as individuals who had no symptoms consistent with COVID-19 during the study period, with or without virological tests).

Reviewer #2 (Comments for the Author):

Please prepare the manuscript using proper format, follow the Journal's guidelines strictly. See the attachment and make the corrections accordingly wherever highlighted red.

Cyclosporine may Reduce the Risk of Symptomatic COVID-19 in Patients with Systemic Lupus Erythematosus: a Retrospective Cohort Study

He-Jun Li^{1,2*#}, Shu-Huan Lin^{2*}, Yan-Qing Wang², Ling Chen², Xiang-Xiong Zheng², Li-Xin Wei^{1,3#}

He-Jun Li, MD; Shu-Huan Lin, MD; Yan-Qing Wang, MD; Ling Chen, MD, PhD; Xiang-Xiong Zheng, MD, Professor; Li-Xin Wei, MD, PhD, Professor.

1 Fujian Institute of Clinical Immunology, Fujian Medical University Union Hospital, Fuzhou, China.

2. Department of Rheumatology, Fujian Medical University Union Hospital, Fuzhou, China.

3. Department of Nephrology, Fujian Medical University Union Hospital, Fuzhou, China.

*He-Jun Li and Shu-Huan Lin contributed equally to this article

#Corresponding to He-Jun Li, MD & Li-Xin Wei, MD, *PhD*, Fujian Medical University Union Hospital, 29 Xinquan Road, Fuzhou 350001, China.

Email: tanklhj@163.com, drwei234@163.com

ABSTRACT

Objectives

This study aimed to explore the effect of cyclosporine (CsA) on SARS-CoV-2 infection in SLE patients to provide a valuable reference for clinical treatment strategies in the context of the long-term risk of SARS-CoV-2 infection.

Methods

SLE patients who visited the rheumatology outpatient department of Fujian Medical University Union Hospital between May 1 and October 31, 2022, were included. Data on SARS-CoV-2 infection in patients between November 1, 2022, and July 31, 2023, were obtained by telephone follow-up. Patients were divided into two groups according to whether CsA was used during the observation period: the GCs or HCQ group and the CsA group. To assess the robustness of the results, Dataset 1 and Dataset 2 were established to be analyzed independently. Multivariate logistic regression was used to estimate ORs and 95% CIs for symptomatic COVID-19.

Results

A total of 184 patients were included, among whom 129 were definite symptomatic COVID-19 patients, 29 were presumptive symptomatic COVID-19 patients, and 4 had signs and symptoms of COVID-19 but tested negative for SARS-CoV-2 using a virological test. According to the multivariable-adjusted models, CsA was associated with lower odds of symptomatic COVID-19 ($P=0.042$, OR= 0.316, 95% CI: 0.104-0.959 in Dataset 1 and $P=0.021$, OR= 0.257, 95% CI: 0.081-0.812 in Dataset 2).

Conclusion

CsA is associated with lower odds of contracting symptomatic COVID-19. The use of CsA may be considered an appropriate therapeutic option for disease management in patients with rheumatic diseases who have severe disease activity and persistent SARS-CoV-2 infection.

IMPORTANCE

Our study indicated that CsA may reduce the risk of symptomatic COVID-19 in SLE patients despite its immunosuppressive effects. This study provides a reference for clinical treatment strategies for AIIRD patients in the context of the long-term risk of SARS-CoV-2 infection.

KEYWORDS: Systemic lupus erythematosus, Cyclosporine, Immunosuppressive therapy, SARS-CoV-2, COVID-19

INTRODUCTION

The coronavirus disease 2019 (COVID-19) pandemic, which is caused by severe acute respiratory syndrome coronavirus 2 (SARS-CoV-2), has had a global effect on public health. Due to the strong infectivity of SARS-CoV-2 and the short duration of immunity after infection, repeated infection with SARS-CoV-2 may be a long-term public health problem. Vaccination against SARS-CoV-2 reduces symptomatic COVID-19 infection rates and poor outcomes (1). Therefore, SARS-CoV-2 infection has little impact on the general population. However, vaccination has additional considerations for people with autoimmune inflammatory rheumatic diseases (AIIRD), including sub-optimal vaccine responses caused by immunomodulatory drugs and rheumatic disease flares (2,3). Moreover, the general recommendation is to postpone vaccination in AIIRD patients with disease activity. Consequently, the overall vaccination rate and effective rate in AIIRD patients are much lower than those in the general population. In addition, given that immunomodulatory drugs widely used in AIIRD patients may be associated with a moderate increase in infection risk, SARS-CoV-2 infection remains a long-standing challenge for AIIRD patients. In the context of the long-term risk of SARS-CoV-2 infection, it is of significant clinical value to explore the impact of various immunomodulatory drugs on the risk of contracting COVID-19 in AIIRD patients.

Cyclosporine (CsA) is an immunomodulatory drug commonly used to treat SLE that can inhibit the activation of T cells. Although its immunosuppressive effects can lead to delayed clearance of the virus, CsA has antiviral effects on cell lines cultured in vitro (4,5). Thus, CsA has dual effects (both beneficial and harmful) on SARS-CoV-2 infection. It is particularly instructive for clinical practice to explore the influence of the dual effects of CsA on SARS-CoV-2 infection in AIIRD patients.

The risk of SARS-CoV-2 infection in AIIRD patients has been controversial in different studies(6,7). The heterogeneity in these results might reflect differences in disease, current levels of disease **activity, and** specific disease-related comorbidities. Moreover, differences in the use of glucocorticoids (GCs) and/or immunomodulatory drugs across studies, both of

which are risk factors for serious infection(8), are another important reason for the inconsistent results. Thus, many confounding factors need to be controlled to explore the real impact of a drug on SARS-CoV-2 infection.

Therefore, this study aimed to explore the effect of CsA on SARS-CoV-2 infection in SLE patients after controlling for confounders to provide a valuable reference for clinical treatment strategies.

PATIENTS AND METHODS

Study cohort and patients

This was a retrospective cohort study of the Chinese Han population. SLE patients who visited the rheumatology outpatient department of Fujian Medical University Union Hospital from May 1 to October 31, 2022, were included. The medical records were retrospectively reviewed, and patients were followed through telephone calls for observation in August 2023. The study included patients 18–75 years of age, inclusive, at the time of screening. The eligibility criteria were as follows: (1) fulfilled either the 2012 Systemic Lupus International Collaborating Clinics (SLICC) Classification Criteria (9) or the 2019 EULAR/ACR Classification Criteria for SLE (10); (2) had a daily maintenance dose of prednisone (or equivalent) ≤ 10 mg during the observation period; (3) received no other immunomodulatory drugs except for GCs, hydroxychloroquine (HCQ) or CsA during the period from May 1 to October 31, 2022, and did not change to different immunomodulatory drugs during the observation period; (4) were not pregnant; (5) did not have COVID-19 before October 31, 2022; and (6) had no irreversible damage to important organs. Patients who were lost to telephone follow-up, were uncertain about infection, or lacked important data would be excluded.

Based on whether CsA was used during the observation period, patients were divided into two groups: the GCs or HCQ group and the CsA group.

Collection and definition of data

We obtained demographic and clinical **data using** a review of electronic medical records and received confirmation from patients during telephone **follow-up**. We analyzed the following parameters: sex, age, clinical manifestations, and rheumatic disease medications taken immediately **before infection**. Information on SLE-related organ involvement, such as cutaneous, arthritis, serositis, lupus nephritis (LN), neuropsychiatric SLE (NPSLE), and hematological involvement (leukopenia, thrombocytopenia, and hemolytic anemia), was collected from the medical records of all included patients.

The study period was from **November 1, 2022** (when the level of COVID-19 prevention and control in China was gradually lowered) to July 31, 2023, and the primary outcome measure was symptomatic COVID-19. Data on SARS-CoV-2 infection among patients were collected through telephone follow-up during the study period. Based on virological detection, patients who had any of the various signs and symptoms of COVID-19 (such as fever, cough, sore throat, malaise, headache, muscle pain, nausea, vomiting, diarrhea, loss of taste and smell) during the study period were categorized as follows: (1) definite symptomatic COVID-19 patients who tested positive for SARS-CoV-2 using a virological test (that is, a nucleic acid amplification test or an antigen test) during the study period; (2) presumptive symptomatic COVID-19 patients who had epidemiological evidence but did not test for virology during the pandemic (the period from November 1, 2022 to February 28, 2023); and (3) non-COVID-19 patients who yielded negative results for SARS-CoV-2 through virological testing.

Asymptomatic patients were defined as individuals who had no symptoms consistent with COVID-19 during the study period, with or without virological tests.

To assess the robustness of the results, we established two datasets—Dataset 1 and Dataset 2—for analysis. In Dataset 1, symptomatic COVID-19 patients included definite and presumptive symptomatic COVID-19 **patients** and non-symptomatic COVID-19 patients included non-COVID-19 patients and asymptomatic patients. In Dataset 2, symptomatic COVID-19 patients referred to definite symptomatic COVID-19 cases, and non-symptomatic COVID-19 patients referred to asymptomatic cases.

Disease duration was defined as the time interval from diagnosis of SLE to October 31, 2022. The observation duration was defined as the time interval between November 1, 2022, and the initial manifestation of any signs or symptoms of COVID-19 for symptomatic patients or until July 31, 2023, for asymptomatic patients.

Statistical analysis

Categorical variables were described as numbers (percentages) and were compared using the **Chi-square test or** Fisher's exact test between the GCs or HCQ group and the CsA group. Normally distributed continuous data were presented as the mean and standard deviation (SD), and differences between groups were tested by one-way analysis of variance.

Abnormally distributed quantitative variables were expressed as medians (interquartile range [IQR]) and were compared between groups using the Mann–Whitney nonparametric U test.

The associations between the risk of symptomatic COVID-19 and baseline characteristics were analyzed by a univariate logistic regression model in Dataset 1 and Dataset 2. To further investigate the effect of CsA on the risk of symptomatic COVID-19, each variable with significant differences ($P < 0.05$) in the univariate logistic regression analysis was included in the multivariable logistic regression model to account for potential confounding factors. **All reported P** values were two-tailed, and statistical significance was defined as $P < 0.05$. All the statistical analyses were performed using R version 4.3.2.

Ethics approval

The cohort study received full ethical approval from the ethics board of Fujian Medical University Union Hospital under project number 2023KY230. Since the study was an observational study conducted through telephone interviews, verbal informed consent was obtained rather than written **informed consent and** general confidentiality principles were abided by.

RESULTS

Study populations

A total of 215 patients were initially enrolled, among whom 18 patients and 4 patients were lost to telephone follow-up in the GCs+HCQ group and the CsA group, respectively. In addition, 4 patients were excluded for missing important data, and 5 patients were excluded for uncertain primary outcomes. Ultimately, 184 patients were included in the analysis. As shown in Table 1, of the 184 patients, 162 patients had one or more signs and symptoms of COVID-19 during the study period. Of these 162 patients, 129 were definite symptomatic COVID-19 patients, 29 were presumptive symptomatic COVID-19 patients, and 4 **were non-COVID-19** patients. There were 2 cases of pneumonia, but no fatalities occurred. The total number of asymptomatic patients was 22, among whom 2 tested positive for SARS-CoV-2.

Comparison of clinical characteristics between the GCs or HCQ group and the CsA group

There was no statistically significant difference in age between the two groups (years, mean \pm SD, 39.6 \pm 12.7 vs. 38.2 \pm 11.2 in Dataset 1 and 39.1 \pm 12.3 vs. 38.2 \pm 10.9 in Dataset 2, respectively, $P > 0.05$). The median observation duration was 2 months (IQR 2-3) in the GCs or HCQ group and 3 months (IQR 2-9) in the CsA group, either in Dataset 1 or Dataset 2. In both datasets, the proportion of people with lupus nephritis (LN) was significantly greater in the CsA group than in the GCs or HCQ group (64.0% vs. 21.4% in Dataset 1 and 68.4% vs. 23.5% in Dataset 2, respectively, $P < 0.001$), while there was no statistically significant difference in other SLE-related organ involvement ($P < 0.05$). The proportion of people with hypertension was greater in the CsA group than in the GCs or HCQ group (32.0% vs. 12.6% in Dataset 1 and 31.6% vs. 9.1% in Dataset 2, respectively, $P < 0.05$). The vaccination rates were greater than 60% in both groups, and there was no statistically significant difference between the two groups. Compared with those in the GCs and HCQ groups, the dose of GCs was slightly greater ($P < 0.05$), while the dose of HCQ tended to decrease in the CsA group ($P = 0.011$ in Dataset 1 and $P = 0.064$ in Dataset 2). In terms of SARS-CoV-2 infection, the

incidence of symptomatic COVID-19 was inclined to decline in the CsA group than in the GCs or HCQ group (68% vs. 88.7%, $P=0.011$ in Dataset 1 and 68.4% vs. 87.9%, $P=0.064$ in Dataset 2). The details are shown in Table 2.

The effect of CsA on the occurrence of symptomatic COVID-19

Univariate logistic regression analysis revealed that CsA ($P=0.009$, odds ratio [OR]= 0.271, 95% CI: 0.103-0.718 in Dataset 1 and $P=0.031$, OR= 0.299, 95% CI: 0.100-0.897 in Dataset 2) reduced the risk of symptomatic COVID-19, and the risk of symptomatic COVID-19 decreased slightly with increasing age ($P=0.012$, OR= 0.958, 95% CI: 0.926-0.991 in Dataset 1 and $P=0.013$, OR= 0.956, 95% CI: 0.915-0.989 in Dataset 2) (Figure 1 and Figure 2). HCQ increased the risk of symptomatic COVID-19 in Dataset 1 ($P=0.007$, OR= 2.501, 95% CI: 1.288-4.856) (Figure 1), while there was no statistically significant effect in Dataset 2 ($P=0.053$, OR= 2.183, 95% CI: 0.990-4.817) (Figure 2). The details are shown in Figure 1 and Figure 2. After adjusting for confounding factors, CsA still showed an effect in reducing the risk of symptomatic COVID-19 ($P=0.042$, OR= 0.316, 95% CI: 0.104-0.959 in Dataset 1 and $P=0.021$, OR= 0.257, 95% CI: 0.081-0.812 in Dataset 2) according to the multivariable logistic regression model (Figure 3). In addition, the multivariable logistic regression model also showed a slight decrease in the risk of symptomatic COVID-19 with increasing age ($P=0.011$, OR= 0.954, 95% CI: 0.919-0.989 in Dataset 1 and $P=0.009$, OR= 0.947, 95% CI: 0.909-0.987 in Dataset 2) (Figure 3).

DISCUSSION

Our study focused on symptomatic COVID-19 as the main endpoint and revealed the impact of CsA on the risk of symptomatic COVID-19 in patients with SLE. To our knowledge, this is the first study to show that CsA may reduce the risk of symptomatic COVID-19 in patients with SLE. Our findings remained largely unchanged, excluding those with a presumptive diagnosis ($n=29$) and those with symptoms but a negative virological test ($n=4$).

Previous studies have focused mainly on the association of immunomodulators with COVID-19-related hospitalization or death(11,12). It is difficult to obtain accurate data on non-

hospitalized COVID-19 patients, as a significant portion of them may remain undocumented. Furthermore, the absence of virological testing led to the underidentification of some presumptive symptomatic COVID-19 cases. The period from November 2022 to February 2023 was the transition stage of China's COVID-19 prevention and control strategy, during which the infection rate of SARS-CoV-2 sharply increased. The short observation duration in our study suggested that most people were infected with SARS-CoV-2 during this period. Meanwhile, after several years of adequate preparation, there was sufficient capacity for virological testing, either nucleic acid amplification by professional institutions or simple antigen testing, to meet the diagnostic requirements of patients. Consequently, most patients (88.0%, 162 out of 184) had one or more signs and symptoms of COVID-19 during the study period, and of these symptomatic patients, the majority (82.1%, 133 out of 162) were tested for SARS-CoV-2 using a virological test, which objectively provided a source of cases with a definite diagnosis for our research. There is, of course, one further point to make. Despite adequate virological test capabilities, 17.9% (29/162) of the enrolled patients with symptoms were not tested for SARS-CoV-2 using a virological test. Moreover, it is important to **note that** no virological test can achieve a 100% positive rate, indicating that the possibility of SARS-CoV-2 infection cannot be completely ruled out even with a negative virological **test result** for SARS-CoV-2. Therefore, to minimize bias and enhance the reliability of our findings, we established two separate datasets for analysis—Dataset 1 and Dataset 2—based on virological detection.

In general, the risk of SARS-CoV-2 infection in people with AIIRDs is affected by immunomodulators, age, disease status, etc. Prednisone doses ≥ 10 mg/day, rituximab, and moderate/high disease activity may increase the risk of COVID-19-related hospitalization(11,13). Therefore, to minimize confounders, a prednisone dose ≤ 10 mg/day was used for patients included in this study, and patients receiving immunomodulatory drugs other than GCs, HCQ, or CsA were excluded. CsA is one of the main drugs used to treat LN, which explains the difference in the proportion of SLE patients with LN and hypertension between the CsA group and the GCs or HCQ group. Moreover, perhaps due to the increased proportion of LN patients in the CsA group, the prednisone dose in this group was slightly

greater than that in the GCs or HCQ group. Despite these differences between the two groups, LN, hypertension, and prednisone dose did not significantly affect the risk of symptomatic COVID-19 according to the univariate logistic regression model ($P > 0.05$), either in Dataset 1 or Dataset 2.

SARS-CoV-2 enters cells by binding to **angiotensin-converting** enzyme 2 (ACE2) through spike proteins and fusing with cell membranes via cleavage of the serine protease TMPRSS2(14). CsA can inhibit the expression of ACE2 in liver cells (15), and molecular docking and all-atom molecular dynamics simulation results revealed strong and stable binding of cyclosporine A (CsA) to the TMPRSS2 gene (16). As a result, CsA may inhibit the pathway by which SARS-CoV-2 enters cells. In addition, cyclophilin A is involved in viral replication (17), and CsA can bind to cyclophilin A and may affect viral replication in the cell. In vitro, CsA dampened viral infection and cytokine release from lung cells upon exposure to three different SARS-CoV-2 variants (4). In this study, multivariate logistic regression models for both Dataset 1 and Dataset 2 showed a reduction in the risk of symptomatic COVID-19 in SLE patients treated with CsA, showing the protective effect of CsA against SARS-CoV-2 in vivo. The consistency of the analysis results between the two datasets indicates the reliability of the research results. Immunosuppressants are commonly used to treat SLE, especially LN. However, the suppression of immunity may lead to a variety of infections in SLE patients. Moreover, SARS-CoV-2 infection may also lead to SLE flares(18). Therefore, in the context of SARS-CoV-2 infection, there is a dilemma regarding the selection of immunosuppressants for controlling disease activity in patients with SLE. Perhaps, due to its potential anti-SARS-CoV-2 effect, CsA could be considered an appropriate treatment option for disease control in this specific scenario, particularly for individuals with severe AIIRD and persistent SARS-CoV-2 infection.

There have been many studies on the role of HCQ in the treatment of COVID-19. One study found that treatment with HCQ was associated with a reduction in COVID-19-associated mortality (19), while another did not (20). Postexposure therapy with HCQ did not prevent SARS-CoV-2 infection or symptomatic COVID-19 in healthy persons exposed to a PCR-

positive patient(21). In our study, a univariate regression model in Dataset 1 but not Dataset 2 showed that an increase in HCQ dose was associated with the risk of symptomatic COVID-19, indicating that HCQ acted as a facilitator rather than a protector against SARS-CoV-2 infection. However, the multivariable logistic regression model showed no significant association between HCQ and the risk of symptomatic COVID-19 ($P > 0.05$). The observed phenomenon may be attributed to the protective effect of CsA against SARS-CoV-2, as the dosage of HCQ was lower in the CsA group than in the GCs and HCQ groups in Dataset 1.

Generally, COVID-19 is more severe in patients older than 65 years. Our results showed that increasing age was associated with a low risk of symptomatic COVID-19. Considering the low average age (less than 40 years) in our study, this approach cannot be applied to the general population.

The following limitations should be noted in the interpretation of our results. The collection of retrospective data is subject to recall bias. To reduce recall bias, we inquired not only about the patient's infection status but also about the infection status of cohabitants to verify each other during the investigation. Besides, it is a single-center study with a small sample size, which could have introduced selection bias.

In conclusion, although CsA has immunosuppressive effects, it may reduce the risk of symptomatic COVID-19 in SLE patients. The use of CsA may be considered an appropriate therapeutic option for disease management in patients with AIIRDs who have severe disease activity and persistent SARS-CoV-2 infection. This study provides a reference for clinical treatment strategies for AIIRD patients in the context of the long-term risk of SARS-CoV-2 infection.

TRANSPARENCY DECLARATION

The authors declare that they have no conflicts of interest.

ACKNOWLEDGEMENTS

We thank all the patients and their families involved in this study.

REFERENCES

1. Fiolet T, Kherabi Y, MacDonald CJ, Ghosn J, Peiffer-Smadja N. Comparing COVID-19 vaccines for their characteristics, efficacy and effectiveness against SARS-CoV-2 and variants of concern: a narrative review. *Clin Microbiol Infect.* 2022 Feb;28(2):202-221. doi: 10.1016/j.cmi.2021.10.005.
2. Zheng YQ, Li HJ, Chen L, Lin SP. Immunogenicity of inactivated COVID-19 vaccine in patients with autoimmune inflammatory rheumatic diseases. *Sci Rep.* 2022 Oct 26;12(1):17955. doi: 10.1038/s41598-022-22839-0.
3. Fan Y, Geng Y, Wang Y, et al. Safety and disease flare of autoimmune inflammatory rheumatic diseases: A large real-world survey on inactivated COVID-19 vaccines. *Ann. Rheum. Dis.* 2022;81(3):443–445. doi: 10.1136/annrheumdis-2021-221736.
4. Fenizia C, Galbiati S, Vanetti C, Vago R, Clerici M, Tacchetti C, et al. Cyclosporine A Inhibits Viral Infection and Release as Well as Cytokine Production in Lung Cells by Three SARS-CoV-2 Variants. *Microbiol Spectr.* 2022 Feb 23;10(1):e0150421. doi: 10.1128/spectrum.01504-21.
5. Wang Y, Li P, Lavrijsen M, Rottier RJ, den Hoed CM, Bruno MJ, et al. Immunosuppressants exert differential effects on pan-coronavirus infection and distinct combinatory antiviral activity with molnupiravir and nirmatrelvir. *United European Gastroenterol J.* 2023 Jun;11(5):431-447. doi: 10.1002/ueg2.12417.
6. Moradi S, Masoumi M, Mohammadi S, Vafaeimanesh J, Mohseni M, Mahdavi H, et al. Prevalence of coronavirus disease 2019 in rheumatic patients and evaluation of the effect of disease-modifying anti-rheumatic drugs. *Intern Emerg Med.* 2021 Jun;16(4):919-923. doi: 10.1007/s11739-020-02535-5.
7. Francesconi P, Cantini F, Profili F, Mannoni A, Bellini B, Benucci M. COVID-19 epidemiology in rheumatic diseases in Tuscany: A case-control study. *Joint Bone Spine.* 2021 May;88(3):105131. doi: 10.1016/j.jbspin.2021.105131.

8. Sepriano A, Kerschbaumer A, Smolen JS, van der Heijde D, Dougados M, van Vollenhoven R, et al. Safety of synthetic and biological DMARDs: a systematic literature review informing the 2019 update of the EULAR recommendations for the management of rheumatoid arthritis. *Ann Rheum Dis.* 2020 Jun;79(6):760-770. doi: 10.1136/annrheumdis-2019-216653.
9. Petri M, Orbai A-M, Alarcón GS, Gordon C, Merrill JT, Fortin PR, et al. Derivation and validation of the Systemic Lupus International Collaborating Clinics classification criteria for systemic lupus erythematosus. *Arthritis Rheum* 2012;64:2677–86. doi: 10.1002/art.34473
10. Aringer M, Costenbader K, Daikh D, et al. 2019 European League Against Rheumatism/American College of Rheumatology Classification Criteria for Systemic Lupus Erythematosus. *Arthritis Rheumatol.* 2019 Sep;71(9):1400-1412. doi: 10.1002/art.40930.
11. Gianfrancesco M, Hyrich KL, Al-Adely S, Carmona L, Danila MI, Gossec L, et al. COVID-19 Global Rheumatology Alliance. Characteristics associated with hospitalisation for COVID-19 in people with rheumatic disease: data from the COVID-19 Global Rheumatology Alliance physician-reported registry. *Ann Rheum Dis.* 2020 Jul;79(7):859-866. doi: 10.1136/annrheumdis-2020-217871.
12. Sparks JA, Wallace ZS, Seet AM, Gianfrancesco MA, Izadi Z, Hyrich KL, et al. COVID-19 Global Rheumatology Alliance. Associations of baseline use of biologic or targeted synthetic DMARDs with COVID-19 severity in rheumatoid arthritis: Results from the COVID-19 Global Rheumatology Alliance physician registry. *Ann Rheum Dis.* 2021 Sep;80(9):1137-1146. doi: 10.1136/annrheumdis-2021-220418.
13. Strangfeld A, Schäfer M, Gianfrancesco MA, Lawson-Tovey S, Liew JW, Ljung L, et al. COVID-19 Global Rheumatology Alliance. Factors associated with COVID-19-related death in people with rheumatic diseases: results from the COVID-19 Global Rheumatology Alliance physician-reported registry. *Ann Rheum Dis.* 2021 Jul;80(7):930-942. doi: 10.1136/annrheumdis-2020-219498.

14. Hoffmann M, Kleine-Weber H, Schroeder S, et al. SARS-CoV-2 Cell Entry Depends on ACE2 and TMPRSS2 and Is Blocked by a Clinically Proven Protease Inhibitor. *Cell*. 2020 Apr 16;181(2):271-280.e8. doi: 10.1016/j.cell.2020.02.052.
15. Niehof M, Borlak J. HNF4alpha dysfunction as a molecular rationale for cyclosporine induced hypertension. *PLoS One*. 2011 Jan 27;6(1):e16319. doi: 10.1371/journal.pone.0016319.
16. Prasad K, Ahamad S, Kanipakam H, et al. Simultaneous Inhibition of SARS-CoV-2 Entry Pathways by Cyclosporine. *ACS Chem Neurosci*. 2021 Mar 3;12(5):930-944. doi: 10.1021/acscchemneuro.1c00019.
17. Watashi K, Shimotohno K. Cyclophilin and viruses: cyclophilin as a cofactor for viral infection and possible anti-viral target. *Drug Target Insights*, 2007, 2: 9–18.
18. Schioppo T, Argolini LM, Sciascia S, Pregnotato F, Tamborini F, Miraglia P, et al. Clinical and peculiar immunological manifestations of SARS-CoV-2 infection in systemic lupus erythematosus patients. *Rheumatology (Oxford)*. 2022 May 5;61(5):1928-1935. doi: 10.1093/rheumatology/keab611.
19. P Arshad S, Kilgore P, Chaudhry ZS, Jacobsen G, Wang DD, Huitsing K, et al. Henry Ford COVID-19 Task Force. Treatment with hydroxychloroquine, azithromycin, and combination in patients hospitalized with COVID-19. *Int J Infect Dis*. 2020 Aug;97:396-403. doi: 10.1016/j.ijid.2020.06.099.
20. RECOVERY Collaborative Group; Horby P, Mafham M, Linsell L, Bell JL, Staplin N, Emberson JR, et al. Effect of Hydroxychloroquine in Hospitalized Patients with Covid-19. *N Engl J Med*. 2020 Nov 19;383(21):2030-2040. doi: 10.1056/NEJMoa2022926.
21. Mitjà O, Corbacho-Monné M, Ubals M, Alemany A, Suñer C, Tebé C, et al. BCN-PEP-CoV2 Research Group. A Cluster-Randomized Trial of Hydroxychloroquine for Prevention of Covid-19. *N Engl J Med*. 2021 Feb 4;384(5):417-427. doi: 10.1056/NEJMoa2021801.

Table 1. Infection and manifestations of SARS-CoV-2 in patients with SLE

	Patients with any of the various signs and symptoms of COVID - 19 (n=162)			Asymptomatic patients (n=22)	
	Definite symptomatic COVID - 19 (n=129)	Presumptive symptomatic COVID - 19 (n=29)	Non - symptomatic COVID - 19 (n=4)	Asymptomatic patients with a positive virological test (n=2)	Asymptomatic patients without a virological test (n=20)
GCs+HCQ group, n (%)	116 (89. 9)	25 (86. 2)	2 (50)	2 (100)	14 (70)
CsA group, n (%)	13 (10. 1)	4 (13. 8)	2 (50)	0	6 (30)
Fever, n (%)	152 (95. 6)	21 (84. 0)	0	0	0
Pneumonia, n (%)	2	0	0	0	0
Death, n (%)	0	0	0	0	0

GCs: glucocorticoids; HCQ:hydroxychloroquine; CsA: Cyclosporine.

Table 2. Comparison of clinical characteristics between GCs or HCQ and CsA group by bivariate analysis in Dataset 1 and Dataset 2

	Dataset 1			Dataset 2		
	GCs or HCQ group (n=159)	CsA group (n=25)	P value	GCs or HCQ group (n=132)	CsA group (n=19)	P value
Gender, F (%)	152 (95.6)	21 (84.0)	0.069	126 (95.5)	16 (84.2)	0.156
Age, years, mean \pm SD	39.6 \pm 12.7	38.2 \pm 11.2	0.603	39.1 \pm 12.3	38.2 \pm 10.9	0.773
Disease duration, years, mean \pm SD	8.5 \pm 5.7	10.3 \pm 6.4	0.147	8.5 \pm 5.7	9.8 \pm 6.7	0.375
Observation duration, months, median [IQR]	2 [2, 3]	3 [2, 9]	0.094	2 [2, 3]	3 [2, 9]	0.052
Mucocutaneous manifestations, n (%)	97 (61.0)	11 (44.0)	0.108	78 (59.1)	8 (42.1)	0.162
Arthritis, n (%)	68 (42.8)	6 (24.0)	0.075	60 (45.5)	5 (26.3)	0.115
Lupus nephritis, n (%)	34 (21.4)	16 (64.0)	<0.001	31 (23.5)	13 (68.4)	<0.001
NPSLE, n (%)	5 (3.1)	1 (4.0)	0.589	3 (2.3)	1 (5.3)	0.419
Hemolytic anemia, n (%)	9 (5.7)	1 (4.0)	1.000	8 (6.1)	0	0.553
Leukopenia, n (%)	38 (23.9)	9 (36.0)	0.197	29 (22.0)	8 (42.1)	0.105
Thrombocytopenia, n (%)	49 (30.8)	6 (24.0)	0.489	38 (28.8)	3 (15.8)	0.234
Serositis, n (%)	11 (6.9)	1 (4.0)	0.910	9 (6.8)	1 (5.3)	1.000
cSLEDAI - 2K, n (%)	0 [0, 0]	0 [0, 0]	0.214	0 [0, 0]	0 [0, 0]	0.119
BMI, median [IQR]	21.5 [19.9, 22.9]	21.8 [19.2, 23.6]	0.862	21.4 [19.8, 22.5]	21.6 [18.7, 23.4]	0.996
Hypertension, n (%)	20 (12.6)	8 (32.0)	0.027	12 (9.1)	6 (31.6)	0.014
Diabetes, n (%)	2 (1.3)	1 (4)	0.356	1 (0.8)	1 (5.3)	0.237
Coronary heart disease, n (%)	0	1 (4.0)	0.136	0	0	
COVID - 19 vaccination, n (%)	118 (74.2)	17 (68.0)	0.514	99 (75.0)	12 (63.2)	0.274
Dose of GCs, mg/d, median [IQR]	2.5 [2.5, 5.0]	5.0 [5.0, 5.0]	<0.001	2.5 [2.5, 5.0]	5.0 [5.0, 5.0]	<0.001
Dose of HCQ, g/d, median [IQR]	0.2 [0.2, 0.2]	0.2 [0.1, 0.2]	0.011	0.2 [0.2, 0.2]	0.2 [0.2, 0.2]	0.064

Dose of						
CsA, mg/d, median	0[0, 0]	150[100, 150]	<0.001	0[0, 0]	150[100, 150]	<0.001
[IQR]						
Symptomatic COVID-19, n (%)	141 (88.7)	17 (68.0)	0.014	116 (87.9)	13 (68.4)	0.057

Disease duration: the time interval from diagnosis of SLE to October 31, 2022.

Observation duration: the time interval from the time interval between November 1, 2022 and the initial manifestation of any signs or symptoms of COVID-19 for symptomatic patients, or until July 31, 2023 for asymptomatic patients.

The dose of GCs, HCQ and CsA referred to the dose taken by patients immediately prior to infection.

NPSLE: neuropsychiatric systemic lupus erythematosus; GCs: glucocorticoids(prednisone); HCQ: hydroxychloroquine; CsA: cyclosporine.

Figure 1 Risk factors for symptomatic COVID-19 in Dataset 1 according to univariate logistic regression analysis. The figure presents the ORs and 95% CIs associated with the end point. OR: odds ratio. See Table 2 note for expansion of additional abbreviations.

Figure 2 Univariate logistic regression analysis of the risk factors for symptomatic COVID-19 in Dataset 2. The figure presents the ORs and 95% CIs associated with the end point. OR: odds ratio. See Table 2 for expansion of additional abbreviations.

Figure 3 Multivariate logistic regression analysis of the effect of CsA on the occurrence of symptomatic COVID-19. The ORs and 95% CIs associated with the end point in (A) Dataset 1 and (B) Dataset 2. OR: odds ratio. See Table 2 note for expansion of additional abbreviations.

Reviewer #1:

First of all, thank you very much for your careful work and detailed advice on our work. It plays a crucial role in improving the quality of our manuscript.

Secondly, we apologize for some unclear expressions. We tried our best to modify the paper according to your valuable comments in detail, so as to improve the readability and quality of the paper. If there are still deficiencies, please do not hesitate to comment.

Best wishes.

Comments for the Author:

The authors appropriately and anonymously declared the number of included/excluded patients in the study. However, please explain clearly how COVID-19 asymptomatic patients without a virological test were recruited for this study, since your statement on Page 5 Para. 3 was confusing (Asymptomatic patients were defined as individuals who had no symptoms consistent with COVID-19 during the study period, with or without virological tests).

Answer:

We sincerely apologize for the confusion our expression has caused you. Our expression can easily lead to misunderstandings, which is caused by our lack of thoughtful consideration. Cyclosporine (CsA) can inhibit the expression of ACE2 and TMPRSS2. As a result, CsA may inhibit the pathway by which SARS-CoV-2 enters cells. Therefore, we assume that CsA may reduce the incidence of SARS-CoV-2 infection. In addition, cyclophilin A is involved in viral replication, and CsA can bind to cyclophilin A and may affect viral replication in the cell. As a result, CsA may reduce the incidence of symptomatic COVID-19. Consequently, in our study, we sought to explore whether the rate of symptomatic infection in SLE patients using CsA was lower than that in SLE patients not using CsA. In other words, could CsA inhibit the infection of SARS-CoV-2, including the onset of infection and symptoms associated with infection? Since the people without symptoms during the study period were not routinely screened for virology, it is difficult to determine whether these people without symptoms were infected with SARS-CoV-2 without symptoms or not infected with SARS-CoV-2.

Therefore, we divided SLE patients into two groups, symptomatic COVID-19

patients and non-symptomatic COVID-19 patients (Page 5 Para. 4). Non-symptomatic COVID-19 patients included “asymptomatic people”. “Asymptomatic people” includes people who were infected with SARS-CoV-2 but had no symptoms, which was what we call asymptomatic infection, as well as people who had no symptoms but it was not clear whether they had been infected with the virus. In the manuscript, “asymptomatic patients” we mentioned actually include these two parts, where “patient” refers to SLE patients, not COVID-19 patients. Your question made us realize that the name “asymptomatic patients” we used was misleading. Therefore, we change the expression to the following: SLE patients without COVID-19-related symptoms were defined as individuals who had no symptoms consistent with COVID-19 during the study period, including individuals who tested positive for SARS-CoV-2 using a virological test, individuals who yielded negative results for SARS-CoV-2 through virological testing, and individuals who had not undergone virus testing. (The original text is as follows: Asymptomatic patients were defined as individuals who had no symptoms consistent with COVID-19 during the study period, with or without virological tests.)

Are our explanations and modifications appropriate? If you have any questions, please feel free to comment.

Thank you again for your comments.

Reviewer #2 (Comments for the Author):

Please prepare the manuscript using proper format, follow the Journal's guidelines strictly. See the attachment and make the corrections accordingly wherever highlighted red.

Answer: First of all, thank you very much for your careful work and detailed revision of our manuscript. It plays a crucial role in improving the quality of our manuscript. We have revised the manuscript according to your comments. If there are still deficiencies, please do not hesitate to comment.

Best wishes.

Re: Spectrum01276-24R1 (Cyclosporine may Reduce the Risk of Symptomatic COVID-19 in Patients with Systemic Lupus Erythematosus: a Retrospective Cohort Study)

Dear Dr. He-Jun Li:

Your manuscript has been accepted, and I am forwarding it to the ASM production staff for publication. Your paper will first be checked to make sure all elements meet the technical requirements. ASM staff will contact you if anything needs to be revised before copyediting and production can begin. Otherwise, you will be notified when your proofs are ready to be viewed.

Sincerely,
Dhammika Navarathna
Editor
Microbiology Spectrum

Reviewer #2 (Comments for the Author):

Authors have addressed most of the comments.